# New Insights on Coding Mutations and mRNA Levels of Candidate Genes Associated with Diarrhea Susceptibility in Baladi Goat

**Mona Al-Sharif [1] and Ahmed Ateya [2,*]**

[1] Department of Biology, College of Science, University of Jeddah, Jeddah 21589, Saudi Arabia
[2] Department of Animal Husbandry and Animal Wealth Development, Faculty of Veterinary Medicine, Mansoura University, Mansoura 35516, Egypt
* Correspondence: ahmed_ismail888@yahoo.com or dr_ahmedismail@mans.edu.eg; Tel.: +2-010-03541921; Fax: +2-050-2372592

**Abstract:** The purpose of this investigation was to examine mutations and mRNA levels of potential genes linked to diarrhea susceptibility in order to assess the health status of diarrheic kids of Baladi goats. One hundred female Baladi kids (35 diarrheic and 65 apparently healthy) were used. PCR-DNA sequencing was conducted for *TMED1, CALR, FBXW9, HS6ST3, SMURF1, KPNA7, FBXL2, PIN1, S1PR5, ICAM1, EDN1, MAPK11, CSF1R, LRRK1,* and *CFH* markers revealed nucleotide sequence variants in the frequency of distribution of all detected SNPs ($p < 0.05$) between healthy and affected kids. Chi-square analysis showed a significant difference between resistant and affected animals. Gene expression profile revealed that *TMED1, CALR, FBXW9, HS6ST3, SMURF1, KPNA7, FBXL2, PIN1, S1PR5, ICAM1, EDN1, MAPK11, CSF1R* and *LRRK1* were significantly up-regulated in diarrheic kids than resistant ones. Meanwhile, *CFH* gene elicited an opposite trend. On the mRNA levels of the examined indicators, there was a substantial interaction between the type of gene and diarrhea resistance/susceptibility. The findings could support the importance of nucleotide variations and the expression pattern of the examined genes as biomarkers for diarrhea resistance/susceptibility and offer a useful management strategy for Baladi goats.

**Keywords:** single nucleotide variants; gene expression profile; diarrhea; Baladi goat

## 1. Introduction

One of the first livestock animals to be domesticated, the domestic goat (*Capra hircus*), is an essential part of animal management [1]. One of the fundamental characteristics of goats is that they have the potential to evolve; for instance, the majority of dairy goat breeds currently in existence are descended from European varieties [2]. There are 4.3 million goats in Egypt. Three goats are raised extensively worldwide, especially in developing countries, where they are exploited as a valuable resource for the manufacture of commodities and a variety of home goods [3–5]. Goat, then, offers the primary way to raise living standards and is essential to both the national economy and the way of life in rural areas [6,7].

The newborn period is a critical stage in the development of physiological processes. During this period, the infant must undergo a number of adjustments to get their bodies ready for life outside the womb [8]. Because of their metabolic instability, newborns are especially susceptible to perinatal illnesses, which have significant fatality rates [8,9]. Neonatal diarrhea is one of the leading causes of illness and mortality in the first three weeks of life and results in considerable economic loss. Its physiological mechanism is that the intestinal mucosa's strong secretion or decreased absorption is caused by a variety of factors [10]. The thin stool is expelled as intestinal peristalsis rises in response to an increase in gut fluid.

The epitheliochorial placenta of ruminants prevents the transmission of immunological components from the mother to the fetus [11]. To obtain efficient passive immunity,

newborn goat calves must be suckled in order to receive immunoglobulin from the does through colostrum. Lambs that are just born are susceptible to infectious diseases, including neonatal diarrhea and respiratory illnesses, until they develop passive immunity through colostrum [12]. Although diarrhea is a defense mechanism of the intestine, it is most frequently brought on by bacteria that inflame the intestinal mucosa and cause it to ooze a lot of mucus, pus, and occasionally even blood [13,14]. It is frequently believed that one or more pathogenic bacteria are to blame for pathogenic diarrhea [15,16]. The gut microbiota is made up of a range of species, including bacteria, viruses, fungi, parasites, and others. It is an important system in both humans and animals [17,18]. The most crucial element of the gut microbiota, which plays several significant roles in digestion, is the intestinal flora [19]. The intestinal flora's bacteria can be classified as probiotics, neutral bacteria, and harmful bacteria depending on how long they spend living in the intestines (long-term colonizers versus temporary passersby), as well as according to how they affect an animal's body (probiotics versus neutral bacteria versus harmful bacteria). The body and the intestinal flora are in a state of mutual restraint. Although pathogenic bacteria play a significant role, it is crucial to consider whether diarrhea is linked to an abnormality in the intestinal flora or is even brought on by the microflora [20].

The candidate gene technique in goat genomics is currently being used to find genetic pathways with various manifestations of growth, reproduction, milk, wool, and disease resistance [21]. There are genotypic and phenotypic variances between various goat populations, according to several genetic characterization attempts of several goat species. As a result, many databases contain genomic data, annotated gene sequences, and gene functions of various goat species [22]. Breeders can identify potential genes and their mutations that result in changes in gene expression and phenotype to uncover genetic markers for specific economic features. Due to the availability of reference whole-genome assemblies of goats, annotated genes, and transcriptomics, comparative genomics is a successful strategy for systemic genetic improvement [23]. Identification and characterization of trait-associated sequence variations and genes will be extremely beneficial for future goat breeding programs [24].

It is significant to mention that additional study is necessary to comprehend potential biomarkers that could signify death and morbidity in newborn goats with diarrhea [25]. Despite the fact that neonatal small ruminant livestock disease and mortality are the subjects of a plethora of scientific knowledge, survival rates have not increased appreciably as a result of this knowledge [26]. The reason may be that neonatal morbidity has not received much attention from researchers, who have instead focused on analyzing and finding solutions to issues brought on by economic considerations [23,25]. In order to improve animal health, it is necessary to look for more accurate possible biomarkers for diagnosing neonatal diseases. In recent investigations, genome-wide association analysis was utilized to identify novel genes specifically for ruminant sensitivity to diarrhea [26,27]. However, no prior research has looked at the SNPs in these genes and their relationship to goat diarrhea susceptibility.

Based on clinical observations in neonatal goats, the molecular changes may clarify the diagnosis of diarrhea and offer important hints about how the physiology of the intestines, inflammation, and immunity are related. As a result, This work used real-time PCR and PCR-DNA sequencing to investigate potential genes (*TMED1, CALR, FBXW9, HS6ST3, SMURF1, KPNA7, FBXL2, PIN1, S1PR5, ICAM1, EDN1, MAPK11, CSF1R, LRRK1,* and *CFH*) impacting the incidence of diarrhea resistance/susceptibility in Baladi goats.

## 2. Materials and Methods

### 2.1. Ethics Statement

The Ethical Committee approved the sample collection and animal care techniques utilized in this study, and they complied with Mansoura University's regulations. The Mansoura University Animal Care and Use Committee (MU-ACUC) gave its approval to the study's protocol (code VM.R.22.11.26).

### 2.2. Animals, Study Design, and Experimental Samples

A total number of 100 female Baladi kids (35 diarrheic and 65 apparently healthy), aged 3 months old and weighing (12.3 ± 2.4 kg), fed whole milk raised at a private farm in Dakahlia province, Egypt, during a period of extended from July to September 2022, were used in this study. During the early stages of the condition, the diarrheic kids were chosen based on a physical examination, paying particular attention to body temperature. The researched kids underwent a clinical examination that included a simultaneous recording of temperature (40.8 ± 0.24), pulse (165 ± 2.6), respiration rates (65 ± 3.45), mucous membranes, and feces characteristics (consistency and color) [28].

Each kid had their jugular veins pierced in order to collect five milliliters of blood. The samples were taken into vacuum tubes containing anticoagulants in order to obtain whole blood for DNA and RNA extraction (EDTA or sodium fluoride).

### 2.3. DNA Extraction and Polymerase Chain Reaction (PCR)

Genomic DNA was isolated from whole blood using the Gene JET whole blood genomic DNA extraction kit and the manufacturer's instructions (Thermo scientific, Vilnius, Lithuania). Using Nanodrop, high-purity and high-concentration DNA was analyzed. By using PCR, segments of coding sites (CDS) for the following *TMED1, CALR, FBXW9, HS6ST3, SMURF1, KPNA7, FBXL2, PIN1, S1PR5, ICAM1, EDN1, MAPK11, CSF1R, LRRK1* and *CFH* genes were amplified. The *Capra hircus* sequence that was published in PubMed served as the basis for the primer sequences. Table 1 provides a list of the primers utilized in the amplification.

With a final volume of 100 μL, the polymerase chain reaction mixture was run in a thermal cycler. Every reaction volume contained 1 μL of each primer, 25 μL of PCR master mix (Jena Bioscience, Hamburg, Germany), 5 μL DNA, and 68 μL $H_2O$ (d.d. water). The first denaturation temperature of 94 °C was applied to the reaction mixture for 8 min. Denaturation at 94 °C for 1 min, annealing at the temperatures listed in Table 1 for 45 s, extension at 72 °C for 45 s, and a final extension at 72 °C for 8 min were all repeated 30 times. Agar gel electrophoresis was used to screen representative PCR analysis results from samples that were kept at 4 °C. A gel documentation system then allowed the fragment pattern to be seen under ultraviolet light.

### 2.4. DNA Sequencing and Polymorphism Detection

Before DNA sequencing, primer dimmers, nonspecific bands, and other contaminants were eliminated. In accordance with the instructions provided by the manufacturer, PCR products of the anticipated size (target bands) were purified using a PCR purification kit (Jena Bioscience # pp-201s/Germany, Jena, Germany), according to Boom et al. [29]. The PCR product was quantified using Nanodrop (Uv-Vis spectrophotometer Q5000/USA) in order to produce high-quality products and ensure appropriate concentrations and purity of the PCR products [30]. PCR results containing the target band were sent for forward and reverse DNA sequencing in order to discover SNPs of the studied genes in diarrhea-free and affected kids.

PCR products were sequenced using an ABI 3730XL DNA sequencer (Applied Biosystems, Waltham, MA, USA) using the enzymatic chain terminator technique developed by Sanger et al. [31]. Chromas 1.45 and BLAST 2.0 programs were used to evaluate the DNA sequencing data [32]. SNPs were discovered as variations between the GenBank reference sequences and the PCR products of the studied genes. Based on a sequence alignment, the MEGA4 program was used to find changes between the enrolled kid's studied genes' amino acid sequences [33].

**Table 1.** Oligonucleotide primers sequence investigated genes utilized in PCR-DNA sequencing.

| Gene | Forward | Reverse | Annealing Temperature (°C) | Length of PCR Product (bp) | Reference |
|---|---|---|---|---|---|
| *TMED1* | 5′-GGAACCGCAACCGGTTAGCAGAC-3′ | 5′-ATCTCCTCAGGCTCCACAGCCT-3′ | 58 | 475 | Current study |
| *CALR* | 5′-TGCAGAGCTGCTGCCGGACGAGT-3′ | 5′-CCTTGTAGTTGAAGATGACATG-3′ | 62 | 517 | Current study |
| *FBXW9* | 5′-TCTGAGGCTGACAGGGCAGGGC-3′ | 5′-CGTCGAGGTAGGCGCAGATCTC-3′ | 58 | 329 | Current study |
| *HS6ST3* | 5′-GGTTCGTGCCGCGCTTCAACTTC-3′ | 5′-AGACCAGTCATCCCCTGGGTAGC-3′ | 60 | 509 | Current study |
| *SMURF1* | 5′-TCGTTGGCGGGAGATGTCGAAC-3′ | 5′- GCTGGCTCCTCCATGAAGCAGCT-3′ | 60 | 552 | Current study |
| *KPNA7* | 5′-TGAGCAGGCCTTGAAGAGGAGGA-3′ | 5′-AGGAGGTAGAAGAGGACGGGCA-3′ | 59 | 618 | Current study |
| *FBXL12* | 5′-TACCTCCAGGTCCGGGATCGGA-3′ | 5′-CCTGCAGGTGCTCGTCGCGGA-3′ | 64 | 647 | Current study |
| *PIN1* | 5′-GAGGAGAAGCTGCCGCCCGGCT-3′ | 5′-ATACTGTGTGACAGGAGAAGGGA-3′ | 58 | 611 | Current study |
| *S1PR5* | 5′-TGAGCGAGGTCATCGTCCAGCA-3′ | 5′-CGCTCCATGGTGAGGAGGCGCTC-3′ | 62 | 385 | Current study |
| *ICAM1* | 5′-GCCAGTCTTAGCCAAGCGCCTC-3′ | 5′- ACCGGACACCTGGCAGCTCAG -3′ | 58 | 469 | Current study |
| *EDN1* | 5′-TCCAAGGAGCTCCAGAAGCAGT-3′ | 5′- CTCCATGGAGTCTTGGTCCTTGA-3′ | 60 | 365 | Current study |
| *MAPK11* | 5′-GCGCGCCGGCTTCTACCGTCTG-3′ | 5′- CGCGCAGCAGCTGGTACACGAG-3′ | 64 | 398 | Current study |
| *CSFIR* | 5′-CGTCAGAGCCAGTGTCTGAGA-3′ | 5′-CTCCAGGCTCAGTGCAGCGGTA-3′ | 60 | 345 | Current study |
| *LRRK1* | 5′-GACTGTTGAGTCGTCCTCTCA-3′ | 5′-CTCCGTCCGTAGCTCCACCAG-3′ | 62 | 530 | Current study |
| *CFH* | 5′-TAGCAGAGGAGAACCTGACACAG-3′ | 5′-ACACTTAACAACTTCACATATG-3′ | 58 | 506 | Current study |

*TMED1* = Transmembrane P24 trafficking protein 1; *CALR* = Calreticulin; *FBXW9* = F-box and WD repeat domain containing 9; *HS6ST3* = Heparan Sulfate 6-O-Sulfotransferase 3; *SMURF1* = SMAD specific E3 ubiquitin protein ligase 1; *KPNA7* = karyopherin subunit alpha 7; *FBXL12* = F-box and leucine rich repeat protein 12; *PIN1* = peptidylprolyl cis/trans isomerase, NIMA-interacting 1; *S1PR5* = sphingosine-1-phosphate receptor 5; *ICAM1* = intercellular adhesion molecule 1; *EDN1* = endothelin 1; *MAPK11* = mitogen-activated protein kinase 11; *CSFIR* = colony stimulating factor 1 receptor; *LRRK1* = leucine rich repeat kinase 1 and *CFH* = complement factor H.

### 2.5. Total RNA Extraction, Reverse Transcription, and Quantitative Real-Time PCR

The manufacturer's recommendations were followed when using Trizol reagent to extract total RNA from kid blood (RNeasy Mini Ki, Catalogue no. 74104). A NanoDrop® ND-1000 Spectrophotometer was used to measure and validate the amount of extracted RNA. Each sample's cDNA was created in accordance with the production methodology (Thermo Fisher, Catalog no, EP0441). SYBR Green PCR Master Mix was utilized in quantitative RT-PCR to ascertain the patterns of gene expression for the studied genes (2x SensiFastTM SYBR, Bioline, CAT No: Bio-98002).

Relative quantification of the amount of mRNA was performed using real-time PCR and SYBR Green PCR Master Mix (Quantitect SYBR green PCR kit, Catalog no, 204141). As stated in Table 2, primer sequences were created in accordance with the PubMed published sequence of *Capra hircus* sequence. The housekeeping gene *ß. actin* was used as a constitutive control for normalization. 25 µL of total RNA, 3 µL of 5x Trans Amp buffer, 4 µL of each primer, 0.25 µL of reverse transcriptase, 12.5 µL of 2x Quantitect SYBR green PCR master mix, and 8.25 µL RNase-free water-made up the reaction mixture.

**Table 2.** Oligonucleotide primers sequence of investigated genes used in real-time PCR.

| Gene | Primer | Product Length (bp) | Annealing Temperature (°C) | Accession Number | Source |
|---|---|---|---|---|---|
| *TMED1* | F5′-CTCCTTCAGCACCATCTCGG-3′<br>R5′-CCCCTTCCTCAAAGGCTCG-3′ | 218 | 60 | XM_005682367.3 | Current study |
| *CALR* | F5-TCGCTGCAAGGACGATGAAT-3′<br>R5′-TTGGCACGATCATCCCAGTC-3′ | 189 | 60 | XM_005682299.3 | Current study |
| *FBXW9* | F5′-CCACAGGTCTCAGATCACGG-3′<br>R5′-ACATTGTGGTGGCTTCGAGT-3′ | 132 | 59 | XM_018051277.1 | Current study |
| *HS6ST3* | F5′-GAAGAAAGACTGTCCCCGCA-3′<br>R5′-TCTGGACGTGTTTCCACTCG-3′ | 107 | 58 | XM_018056433.1 | Current study |
| *SMURF1* | F5′-AAGGCCATACCTCTGAGCCC-3′<br>R5′-GTGGTGTAACCGTGGGTCTG-3′ | 161 | 62 | XM_018040226.1 | Current study |
| *KPNA7* | F5′-CTGAGGGCATTTAAGGCCGA-3′<br>R5-GATGCATCCTTGCCCCGATA-3′ | 140 | 60 | XM_005697882.3 | Current study |
| *FBXL12* | F5′-CCACGATGTCAGAAACCAACG-3′<br>R5′-TGGGAGCCTGAGAACAGGTA-3′ | 126 | 59 | XM_005682343.2 | Current study |
| *PIN1* | F5′-TCACAGATTCGGGCATCCAC-3′<br>R5′-CCAGCCATTCTGGGGTCAAT-3′ | 191 | 59 | XM_018051006.1 | Current study |
| *S1PR5* | F5′-TACACAGGCTGCGAACCATT-3′<br>R5′-GACTCACAGCAAGTCACGGA-3′ | 128 | 60 | XM_018051062.1 | Current study |
| *ICAM1* | F5′-GTGGGACCACACAGTTCCAA-3′<br>R5′-GGACATGAAACCTCGCCTCA-3′ | 166 | 60 | XM_005682354.3 | Current study |
| *EDN1* | F5′-TTGAGATCCGGAGAACCCGA-3′<br>R5′-GGGCGGATTAAAGGAAGGGT-3′ | 137 | 58 | XM_005696851.3 | Current study |
| *MAPK11* | F5-CCGTCTGGAGCTGAACAAGA-3′<br>R5′-GGATCAGAGACTGGAAGGGC-3′ | 164 | 59 | XM_018048955.1 | Current study |
| *CSFIR* | F5-TACTCCTTCTCGCCGTGGTA-3′<br>R5′-GGATCAGAGACTGGAAGGGC-3′ | 193 | 60 | XM_018050168.1 | Current study |
| *LRRK1* | F5-CCGTGTCCTCATCCCTTGAG-3′<br>R5′-AAGTCAGTTCCGTCGTGGTC-3′ | 187 | 60 | XM_013972914.2 | Current study |
| *CFH* | F5-ATTGCTGGGGCTCCTGACT-3′<br>R5′-AGTGCAGGAAACCACTTGCT-3′ | 130 | 62 | XM_018060122.1 | Current study |
| *ß. actin* | F5′-ATGTATGTGGCCATCCAGGC-3′<br>R5′-TGAGGTAGTCCGTCAGGTCC-3′ | 175 | 58 | AF481159.1 | Current study |

*TMED1* = Transmembrane P24 trafficking protein 1; *CALR* = Calreticulin; *FBXW9* = F-box and WD repeat domain containing 9; *HS6ST3* = Heparan Sulfate 6-O-Sulfotransferase 3; *SMURF1* = SMAD specific E3 ubiquitin protein ligase 1; *KPNA7* = karyopherin subunit alpha 7; *FBXL12* = F-box and leucine rich repeat protein 12; *PIN1* = peptidylprolyl cis/trans isomerase, NIMA-interacting 1; *S1PR5* = sphingosine-1-phosphate receptor 5; *ICAM1* = intercellular adhesion molecule 1; *EDN1* = endothelin 1; *MAPK11* = mitogen-activated protein kinase 11; *CSFIR* = colony stimulating factor 1 receptor; *LRRK1* = leucine rich repeat kinase 1 and *CFH* = complement factor H.

The finished reaction mixture was placed in a thermal cycler and put through the program listed below: reverse transcription at 50 °C for 30 min, initial denaturation at 94 °C for 10 min, followed by 40 cycles of 94 °C for 15 s, annealing temperatures as listed in Table 2, and extension at 72 °C for 30 s. After the amplification phase, a melting curve analysis was performed to confirm the specificity of the PCR product. The relative expression of each gene in each sample in proportion to the *ß. actin* gene was measured and determined using the $2^{-\Delta\Delta Ct}$ method [34,35].

### 2.6. Statistical Analysis

Ho: Coding mutations and mRNA levels of *TMED1*, *CALR*, *FBXW9*, *HS6ST3*, *SMURF1*, *KPNA7*, *FBXL2*, *PIN1*, *S1PR5*, *ICAM1*, *EDN1*, *MAPK11*, *CSF1R*, *LRRK1* and *CFH* markers are not related with diarrhea susceptibility in Baladi goat.

HA: Coding mutations and mRNA levels of *TMED1*, *CALR*, *FBXW9*, *HS6ST3*, *SMURF1*, *KPNA7*, *FBXL2*, *PIN1*, *S1PR5*, *ICAM1*, *EDN1*, *MAPK11*, *CSF1R*, *LRRK1* and *CFH* markers are related with diarrhea susceptibility in Baladi goat.

The statistical analysis was conducted using the Graphpad statistics program (Graphpad prism for Windows version 5.1, Graphpad Software, Inc., San Diego, CA, USA). The Chi-square test was used to examine the distribution of the discovered SNPs between the two groups using (Crosstabs). Statistical analysis was used to examine whether there were differences in the frequency of gene SNPs between diarrhea-affected and resistant kids ($p < 0.05$).

The Statistical Package for Social Science (SPSS) version 17 computer program and the $t$-test were used to determine the statistical significance of the difference between diarrheal and healthy Baladi kids (SPSS Inc, Chicago, IL, USA). The results were shown as mean± standard error (Mean ± SE). At $p < 0.05$, differences were deemed significant. A discriminant analysis model was used to assess the significance of numerous variables in order to discriminate between sick and healthy goats as a dependent variable using the gene expression profile of the investigated genes as an independent variable. The mRNA levels of the genes under investigation were used as the basis for the distinction between diarrheal and healthy children. A univariate general linear model (GLM) with the two-way ANOVA was used to assess the interaction between two components (gene type and diarrhea resistance/susceptibility) and its effects on the gene expression outcomes parameter.

## 3. Results

### 3.1. PCR-DNA Sequencing of Candidate Genes

Nucleotide sequence variation in the form of SNPs related to diarrhea was discovered between healthy and Baladi kids using the results of PCR-DNA sequencing of the *TMED1* (475-bp), *CALR* (517-bp), *FBXW9* (329-bp), *HS6ST3* (509-bp), *SMURF* (552-bp), *KPNA7* (618-bp), *FBXL2* (647-bp), *PIN1* (611-bp), *S1PR5* (385-bp), *ICAM1* (469-bp), *EDN1* (365-bp), *MAPK11* (398-bp), *CSF1R* (345-bp), *LRRK1* (530-bp) and *CFH* (506-bp) genes. Nucleotide sequence variations between the examined genes in the research animals and the reference sequences retrieved in GenBank were used to validate all discovered SNPs (Supplementary Figures S1–S15). Diarrhea-healthy and affected kids had significantly different frequencies of the examined genes, according to chi-square analysis of discovered SNPs ($p < 0.05$) (Table 3). The variants identified in Table 3 are all located within the exonic region of studied genes, resulting in coding mutations between healthy and diarrheic kids.

**Table 3.** SNP distribution and kind of mutation for the genes under investigation in diarrheal and healthy Baladi kids.

| Gene | SNPs | Healthy *n = 65* | Diarrheic *n = 35* | Total *n = 100* | Type of Mutation | Amino Acid Number and Type | Chi Value |
|---|---|---|---|---|---|---|---|
| *TMED1* | T74G | 29 | - | 29/100 | Non-synonymous | 25 F to C | 73.11 |
| | G260C | - | 23 | 23/100 | Non-synonymous | 87 W to S | 57.98 |
| *CALR* | G147A | - | 18 | 18/100 | Synonymous | 49 L | 45.38 |
| | C312T | 34 | - | 34/100 | Synonymous | 104 P | 85.72 |
| | C384T | 26 | - | 26/100 | Synonymous | 128 G | 65.55 |
| | G465A | 14 | - | 14/100 | Synonymous | 155 P | 35.29 |
| *FBXW9* | C190A | 37 | - | 37/100 | Non-synonymous | 64 R to S | 93.28 |

**Table 3.** *Cont.*

| Gene | SNPs | Healthy *n* = 65 | Diarrheic *n* = 35 | Total *n* = 100 | Type of Mutation | Amino Acid Number and Type | Chi Value |
|---|---|---|---|---|---|---|---|
| *HS6ST3* | T80C | - | 22 | 22/100 | Non-synonymous | 27 L to S | 55.46 |
| | A212G | 17 | - | 17/100 | Non-synonymous | 71 K to R | 42.86 |
| | A470G | 41 | - | 41/100 | Non-synonymous | 157 Q to R | 103.35 |
| *SMURF1* | T322C | - | 25 | 25/100 | Non-synonymous | 10 C to R | 63.03 |
| *KPNA7* | A83C | 36 | - | 36/100 | Non-synonymous | 28 H to P | 90.74 |
| | C202T | 15 | - | 15/100 | Non-synonymous | 68 R to C | 37.81 |
| | G307A | - | 13 | 13/100 | Non-synonymous | 103 G to R | 32.77 |
| | A457G | 37 | - | 37/100 | Non-synonymous | 153 K to E | 93.28 |
| *FBXL12* | G285A | - | 19 | 19/100 | Synonymous | 95 T | 47.90 |
| | A580G | - | 21 | 21/100 | Non-synonymous | 194 S to G | 52.94 |
| *PIN* | T81C | 16 | - | 16/100 | Synonymous | 27 N | 40.34 |
| | T132C | - | 19 | 19/100 | Synonymous | 44 N | 47.90 |
| | A198G | - | 27 | 27/100 | Synonymous | 66 R | 68.06 |
| | A447G | 43 | - | 43/100 | Synonymous | 149 T | 108.41 |
| *S1PR5* | C197A | - | 13 | 13/100 | Non-synonymous | 66 A to E | 32.77 |
| | A272G | 31 | - | 31/100 | Non-synonymous | 91 Y to C | 78.15 |
| *ICAM1* | G53C | 46 | - | 46/100 | Non-synonymous | 18 C to S | 115.96 |
| | G118C | 22 | - | 22/100 | Non-synonymous | 40 A to P | 55.46 |
| | G226A | 54 | - | 54/100 | Non-synonymous | 76 V to I | 136.13 |
| *EDN1* | C73T | - | 29 | 29/100 | Non-synonymous | 25 F to L | 73.11 |
| *MAPK11* | G109A | - | 15 | 15/100 | Non-synonymous | 37 G to S | 37.81 |
| | T265G | - | 30 | 30/100 | Non-synonymous | 89 C to G | 75.63 |
| | G370C | 49 | - | 49/100 | Non-synonymous | 124 A to P | 123.53 |
| *CSFIR* | C63T | 37 | - | 37/100 | Synonymous | 21 A | 93.28 |
| | G143A | - | 26 | 26/100 | Non-synonymous | 48 R to Q | 65.55 |
| | C284T | 19 | - | 19/100 | Non-synonymous | 95 S to F | 47.90 |
| *LRRK1* | C53T | - | 23 | 23/100 | Non-synonymous | 18 T to M | 57.98 |
| | G85C | - | 32 | 32/100 | Non-synonymous | 29 V to L | 80.67 |
| | T174C | 24 | - | 24/100 | Synonymous | 58 G | 60.50 |
| | T337C | 48 | - | 48/100 | Non-synonymous | 113 S to P | 121.01 |
| | G370A | - | 28 | 28/100 | Non-synonymous | 124 E to K | 70.59 |
| *CFH* | G95C | 33 | - | 33/100 | Non-synonymous | 32 G to A | 83.19 |
| | C458A | 21 | - | 21/100 | Non-synonymous | 153 T to K | 52.94 |

*TMED1* = Transmembrane P24 trafficking protein 1; *CALR* = Calreticulin; *FBXW9* = F-box and WD repeat domain containing 9; *HS6ST3* = Heparan Sulfate 6-O-Sulfotransferase 3; *SMURF1* = SMAD specific E3 ubiquitin protein ligase 1; *KPNA7* = karyopherin subunit alpha 7; *FBXL12* = F-box and leucine rich repeat protein 12; *PIN1* = peptidylprolyl cis/trans isomerase, NIMA-interacting 1; *S1PR5* = sphingosine-1-phosphate receptor 5; *ICAM1* = intercellular adhesion molecule 1; *EDN1* = endothelin 1; *MAPK11* = mitogen-activated protein kinase 11; *CSFIR* = colony stimulating factor 1 receptor; *LRRK1* = leucine rich repeat kinase 1 and *CFH* = complement factor H. A = Alanine; C = Cisteine; D = Aspartic acid; E = Glutamic acid; F = Phenylalanine; G = Glycine; H = Histidine; I = Isoleucine = K = Lysine; L = Leucine; M = Methionine; N = Asparagine; P = Proline; Q = Glutamine; R = Argnine; S = Serine; T = Threonine; and V = Valine.

### 3.2. Gene Expression Pattern

Figure 1 shows the gene expression profile of the tested markers. Kids with diarrhea expressed significantly higher levels of the genes *TMED1*, *CALR*, *FBXW9*, *HS6ST3*, *SMURF1*, *KPNA7*, *FBXL2*, *PIN1*, *S1PR5*, *ICAM1*, *EDN1*, *MAPK11*, *CSF1R*, and *LRRK1*. *CFH* gene was significantly down-regulation in diarrheic kids than resistant ones.

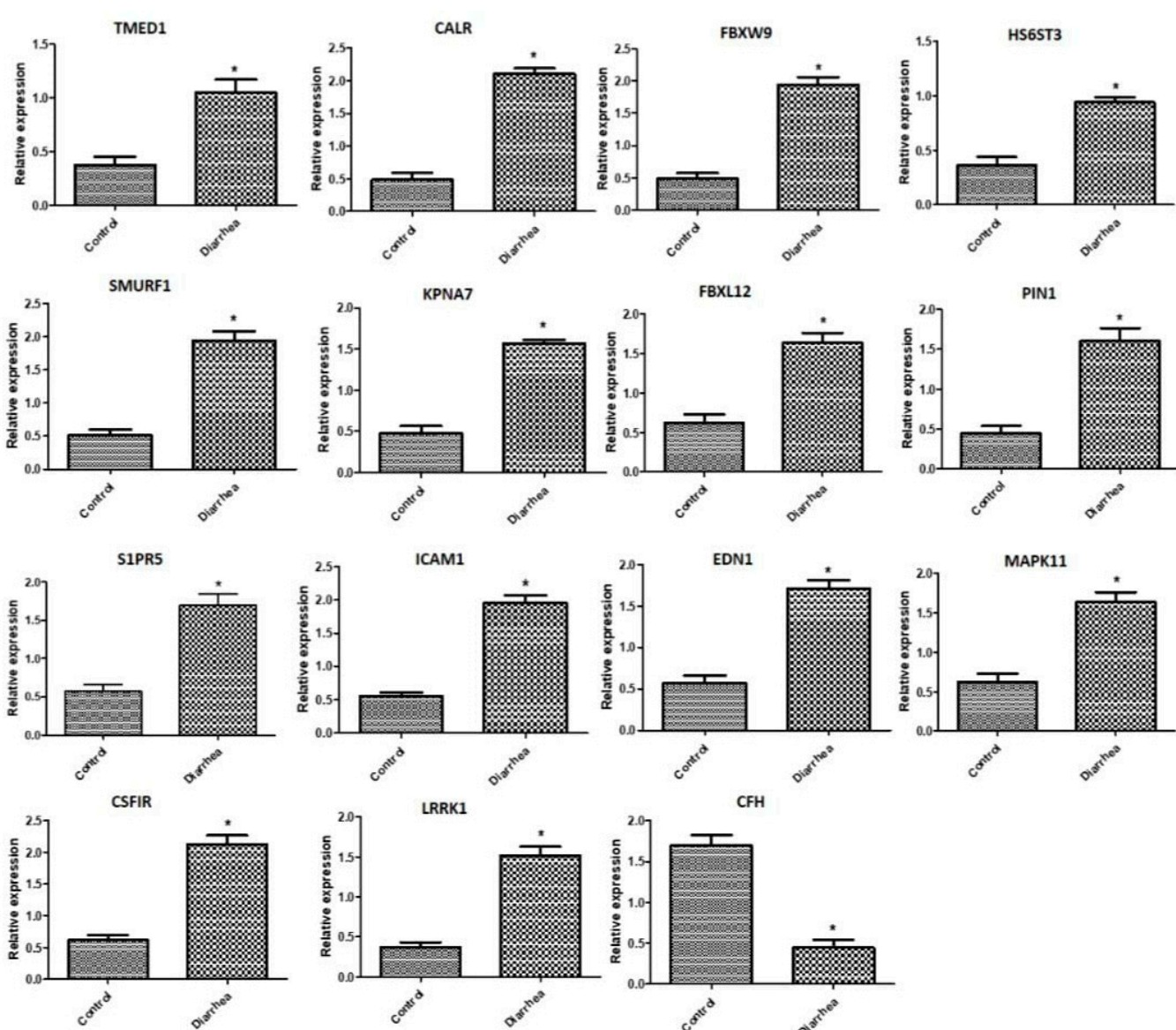

**Figure 1.** Comparative expression patterns of the genes *TMED1*, *CALR*, *FBXW9*, *HS6ST3*, *SMURF1*, *KPNA7*, *FBXL2*, *PIN1*, *S1PR5*, *ICAM1*, *EDN1*, *MAPK11*, *CSF1R*, *LRRK1*, and *CFH* in healthy and diarrhea affected Baladi kids. * means significance when $p < 0.05$.

On the mRNA levels of the examined indicators, there was a substantial interaction between the type of gene and diarrhea resistance/susceptibility. The greatest potential level of mRNA for each gene assessed in the diarrheic kids was found for *CSIRF* (2.12 ± 0.13); the lowest level was found for *CFH* (0.45 ± 0.09). The greatest potential level of mRNA among all the genes examined in the healthy kids was found for *CFH* (1.69 ± 0.13), while the lowest level was found for *HS6ST3* (0.34 ± 0.07).

## 4. Discussion

The most prevalent disease in the newborn stage is diarrhea, which is typically attributed to infectious sources such as bacteria, viruses, and protozoa [36]. Increased newborn diarrhea was linked to wet spring and summer weather as well as underdeveloped immunological function [37]. Neonatal diarrhea mortality rates varied according to management and conditions for preventing epidemics.

Farm animal species' genetic makeup includes a diverse range of genetic variants that can have positive or negative effects on health and productivity. Single nucleotide polymorphisms (SNPs) are one type of these differences [38]. The majority of cases are caused by SNPs that have varying degrees of effects on gene action, including the substitution of one amino acid for another, duplications and deletions that result in a frameshift and premature termination of translation, and complete deletion of entire exon(s) or gene(s) in affected individuals. It is known that these modifications in the coding areas affect how mRNA splicing patterns behave or how proteins function [39]. In this study, a PCR-DNA sequencing technique was used to characterize the *TMED1*, *CALR*, *FBXW9*, *HS6ST3*, *SMURF1*, *KPNA7*, *FBXL2*, *PIN1*, *S1PR5*, *ICAM1*, *EDN1*, *MAPK11*, *CSF1R*, *LRRK1*, and *CFH* genes in diarrhea healthy and affected Baladi kids at the molecular level. The findings showed that there were SNPs between the two groups. Chi-square analysis demonstrated significance in the SNP distribution among the examined kids. It is noteworthy to note that the polymorphisms discovered and published here disclose additional information about the examined genes when compared to the appropriate GenBank reference sequence.

Recent studies conducted the genome-wide association analysis to target new genes specific for diarrhea susceptibility in livestock [26,27]; however, until now, there are no previous studies investigating these genes' SNPs and their association with diarrhea susceptibility. With the help of the goat (*Capra hircus*) gene sequences published in PubMed, our study is the first to indicate this association. Additionally, to the best of our knowledge, no research has previously examined the polymorphism of *TMED1*, *CALR*, *FBXW9*, *HS6ST3*, *SMURF1*, *KPNA7*, *FBXL2*, *PIN1*, *S1PR5*, *ICAM1*, *EDN1*, *MAPK11*, *CSF1R*, *LRRK1*, and *CFH* genes and their relationship to diarrhea in goat. However, the candidate gene approach was used for monitoring the health status of diarrheic newly born animals. For instance, dairy calves showed no discernible connection between *CXCR1* SNPs and clinical intestinal disorders [40]. Piglet diarrhea has also been linked to genetic variation in the *DRA* gene for swine leukocyte antigen [41]. Additionally, research on the Nramp1 gene polymorphism and its connection to diarrhea in pigs was conducted [42].

According to the idea of genetic genomics, transcript abundance works as a heritable endophenotype, and genomic polymorphisms are linked to it [43]. This method supports the idea that combining data on gene expression and chromosomal variants could aid in our understanding of the genetics underlying the onset of disease [44]. Quantitative trait loci (QTLs) for expression are polymorphisms connected to gene expression [45]. In the current study, we proposed that an individual's genetic variability in the transcriptional response to diarrhea susceptibility may have an impact on the disease's course. *TMED1*, *CALR*, *FBXW9*, *HS6ST3*, *SMURF1*, *KPNA7*, *FBXL2*, *PIN1*, *S1PR5*, *ICAM1*, *EDN1*, *MAPK11*, *CSF1R*, *LRRK1*, and *CFH* genes were quantified using real-time PCR in both diarrhea-resistant and non-resistant Baladi kids. According to our research, resistant Baladi kids had lower levels of the expression of the genes *TMED1*, *CALR*, *FBXW9*, *HS6ST3*, *SMURF1*, *KPNA7*, *FBXL2*, *PIN1*, *S1PR5*, *ICAM1*, *EDN1*, *MAPK11*, *CSF1R*, and *LRRK1* genes than diarrheic ones. However, the *CFH* gene showed a different trend. Our study is the first to use a real-time PCR technique to identify mRNA levels of these markers in resistant and susceptible kids to diarrhea.

Using RFLP and SNP genetic markers in domestic animals, prior research investigated the polymorphism of immune genes in ruminants [40–42]. Contrarily, we used SNP genetic markers and gene expression to examine gene polymorphism in order to overcome the limitations of previous studies. As a result, in both diarrheal and healthy kids, the examined gene regulation mechanisms are well understood. As far as we are aware, there

aren't many reports on the gene expression profile of immunological markers connected to cow diarrhea susceptibility. When compared to the gene expression profile in ruminants, a transcriptase study of goat peripheral blood mononuclear cells (PBMCs) infected with bovine viral diarrhea virus-2 demonstrated distinct immune-related gene expression [22]. In addition, healthy and diarrheal neonatal goats both have identical TLR4 and its downstream signaling pathways [25].

The family of transmembrane proteins with the emp24 domain, which is involved in the movement of proteins across vesicles, includes transmembrane P24 trafficking protein 1 (TMED1) [46]. Due to its interaction with the innate immune system protein interleukin 1 receptor-like 1 (IL1RL1), the protein that this gene encodes has been identified [46]. The calreticulin (*CALR*) gene encodes a multifunctional protein that helps to keep the proper concentrations of calcium ions in this structure [47]. Calreticulin is hypothesized to influence gene activity, cell proliferation, migration, adhesion, and other processes through calcium regulation and other mechanisms [48]. It also regulates apoptosis, which is the process of controlled cell death. This protein is essential for both the immune system and wound healing [47].

The F-box protein family, which includes proteins such as F-box and WD repeat domain containing 9 (FBXW9) and F-box and leucine-rich repeat protein, is characterized by a 40-amino acid F-box motif (FBXL12). F-box proteins serve as protein-ubiquitin ligases [49]. Through additional protein interaction domains, F-box proteins communicate with ubiquitination targets [49]. The innate immune system and Class I MHC-mediated antigen processing and presentation are two of the pathways that the protein-coding genes FBXW9 and FBXL12 are linked with [50]. Heparan sulfate (HS) sulfotransferases, including HS6ST3, alter HS to produce the structures necessary for interactions between HS and other proteins [51]. Proliferation and differentiation, adhesion, migration, inflammation, and blood coagulation are all affected by these interactions [51].

Through the bone morphogenetic signaling system, cell motility, polarity, and signaling are regulated by SMAD-specific E3 ubiquitin protein ligase 1 (SMURF1) [52]. Inflammatory Bowel Disease 24 and Brachydactyly, Type A2 are diseases linked to SMURF1 [53]. Karyopherin subunit 7 (KPNA7) helps build the mitotic spindle, which directs the duplicated chromosomes during mitosis, and is involved in the import of nuclear proteins, both of which improve the accuracy of cell division [54]. It is widely known that Peptidylprolyl cis/trans isomerase, NIMA-interacting 1 (PIN1) affects key proteins involved in the human immune response [55]. Sphingosine 1-phosphate receptor 5 (S1PR5) is a plausible candidate gene because of its function as a sphingosine 1-phosphate receptor that regulates apoptosis [56]. Despite the fact that S1PR5 has no known role in T cells, the natural killer (NK) cell migration that S1PR5 promotes enhances NK cell egress from the bone marrow (BM) and LNs [57].

The *ICAM1* gene produces the protein known as CD54 (Cluster of Differentiation 54), also known as intercellular adhesion molecule 1 (ICAM-1) [58]. On immune system cells and endothelial cells, this gene typically expresses a cell surface glycoprotein. ICAM-1 participates in inflammatory processes and the T cell-mediated host defense mechanism [58]. It functions as a co-stimulatory molecule on antigen-presenting cells to activate MHC class II-restricted T-cells, while on other cell types, it collaborates with MHC class I to activate cytotoxic T-cells [58]. Endothelial cells primarily release endothelin-1 (ET-1), an effective endogenous vasoconstrictor [59]. The inflammatory marker endothelin1 (EDN1) is well known [59]. EDN1's effects are primarily mediated by endothelin receptors of type A (EDNRA) [60]. ET-1 synthesis and release are induced by angiotensin II (AII), antidiuretic hormone (ADH), thrombin, cytokines, reactive oxygen species, and shearing forces acting on the vascular endothelium [60].

Inflammatory diseases may be treated by targeting mitogen-activated protein kinases (MAPKs), which are typically activated in response to inflammatory cytokines and cellular stress [61]. Colony stimulating factor 1 receptor (CSF1R), also known as macrophage colony-stimulating factor receptor (M-CSFR) and CD115 (Cluster of Differentiation 115), is a protein

produced by this gene that controls the differentiation and function of macrophages [62]. Leucine-rich repeat kinase 1 (LRRK1) has a limited impact on bone production parameters but is essential in controlling cytoskeletal architecture, osteoclast activity, and bone resorption [63]. In the mouse spleen, B cells and macrophages both express LRRK2 protein, and one of the B cell subsets, B2 cells, express *LRRK2* mRNA at a considerably higher level than B1 cells [64]. These results imply that LRRK2 plays significant functions in the immune system as well as the neurological system. The complement factor H (*CFH*) genes control the complement system, a component of the body's immune response that destroys foreign invaders, causes an inflammatory reaction and clears debris from cells and tissues [65]. Complement factor H (CFH) protects healthy cells by preventing the complement system from becoming overactive (activated). Increased levels of inflammatory mediators and different amounts of complement proteins are brought on by the loss of the factor H protein (FH) [66]. The latter discovery could explain its substantial down-regulation in diarrheic kids.

The marked alteration in the expression pattern of *TMED1*, *CALR*, *FBXW9*, *HS6ST3*, *SMURF1*, *KPNA7*, *FBXL2*, *PIN1*, *S1PR5*, *ICAM1*, *EDN1*, *MAPK11*, *CSF1R*, *LRRK1*, and *CFH* markers in diarrheic kids may be attributed to the fact that the intestine is the primary site for the presence of numerous bacteria, nutrients, and interactions between immune cells; therefore, it is more sensitive to peroxidation than healthy tissues [67–70]. The invasion of gastrointestinal pathogens may be a strong oxidizing stimulus, which triggers immune responses to deal with the pathogen attack by activating neutrophils and macrophage activity. This causes an excessive amount of ROS to be produced and to build up, which ultimately causes oxidative stress [71]. Because they significantly alter the mechanism regulating gut barrier function and may increase intestinal permeability to pathogenic bacteria, TLR4, and its downstream signaling pathways are obviously crucial for inducing the release of inflammatory cytokines during bacterial infection and diarrhea [72–74]. Recent studies that found that the rise of the E. coli population was accompanied by an increase in the concentration of ROS in the intestine [75,76] indicated that ROS is also involved in the inter-microbial competition. This research implies that oxidative stress plays a significant part in the emergence of infectious diseases. We, therefore, presume that the neonatal diarrheic kids in this study are affected by an infectious etiology. Our Real-Time PCR data also provide compelling proof that the diarrheal kids were experiencing a significant inflammatory response.

## 5. Conclusions

Single nucleotide polymorphisms (SNPs) related to diarrhea resistance/susceptibility were discovered between diarrhea healthy and affected Baladi kids by PCR-DNA sequencing of *TMED1*, *CALR*, *FBXW9*, *HS6ST3*, *SMURF1*, *KPNA7*, *FBXL2*, *PIN1*, *S1PR5*, *ICAM1*, *EDN1*, *MAPK11*, *CSF1R*, *LRRK1*, and *CFH* genes. Furthermore, mRNA levels of these markers varied between healthy and affected kids. These distinctive functional variants provide a promising opportunity to lessen goat diarrhea by selective breeding of animals utilizing genetic markers connected to natural resistance. Variable gene expression patterns in resistant and susceptible kids to diarrhea may serve as a guide and a biomarker for monitoring the health of kids. The gene targets identified here may also facilitate future treatment approaches for diarrhea.

**Supplementary Materials:** The following supporting information can be downloaded at: https://www.mdpi.com/article/10.3390/agriculture13010143/s1, Figure S1: *TMED1* gene (475-bp) demonstrative DNA sequence alignment between healthy and diarrheal Baladi kids, together with the reference sequence found in GenBank (gb|XM 005682367.3); Figure S2: *CALR* gene (517-bp) demonstrative DNA sequence alignment between healthy and diarrheal Baladi kids, together with the reference sequence found in GenBank gb|XM_005682299.3|; Figure S3: *FBXW9* gene (329-bp) demonstrative DNA sequence alignment between healthy and diarrheal Baladi kids, together with the reference sequence found in GenBank gb|XM_018051277.1|; Figure S4: *HS6ST3* gene (509-bp) demonstrative DNA sequence alignment between healthy and diarrheal Baladi kids, together with the reference

sequence found in GenBank gb|XM_018056433.1|; Figure S5: *SMURF1* gene (552-bp) demonstrative DNA sequence alignment between healthy and diarrheal Baladi kids, together with the reference sequence found in GenBank gb|XM_018040226.1|; Figure S6: *KPNA7* gene (618-bp) demonstrative DNA sequence alignment between healthy and diarrheal Baladi kids, together with the reference sequence found in GenBank gb|XM_005697882.3|; Figure S7: *FBXL12* gene (647-bp) demonstrative DNA sequence alignment between healthy and diarrheal Baladi kids, together with the reference sequence found in GenBank gb|XM_005682343.2|; Figure S8: *PIN1* gene (611-bp) demonstrative DNA sequence alignment between healthy and diarrheal Baladi kids, together with the reference sequence found in GenBank gb|XM_018051006.1|; Figure S9: *S1PR5* gene (385-bp) demonstrative DNA sequence alignment between healthy and diarrheal Baladi kids, together with the reference sequence found in GenBank gb|XM_018051062.1|; Figure S10: *ICAM1* gene (469-bp) demonstrative DNA sequence alignment between healthy and diarrheal Baladi kids, together with the reference sequence found in GenBank gb|XM_005682354.3|; Figure S11: *EDN1* gene (365-bp) demonstrative DNA sequence alignment between healthy and diarrheal Baladi kids, together with the reference sequence found in GenBank gb|XM_005696851.3|; Figure S12: *MAPK11* gene (398-bp) demonstrative DNA sequence alignment between healthy and diarrheal Baladi kids, together with the reference sequence found in GenBank gb|XM_018048955.1|; Figure S13: *CSFIR* gene (345-bp) demonstrative DNA sequence alignment between healthy and diarrheal Baladi kids, together with the reference sequence found in GenBank gb|XM_018050168.1|; Figure S14: *LRRK1* gene (530-bp) demonstrative DNA sequence alignment between healthy and diarrheal Baladi kids, together with the reference sequence found in GenBank gb|XM_013972914.2|; Figure S15: *CFH* gene (506-bp) demonstrative DNA sequence alignment between healthy and diarrheal Baladi kids, together with the reference sequence found in GenBank gb|XM_018060122.1|.

**Author Contributions:** Conceptualization, A.A.; methodology, A.A. and M.A.-S.; validation, A.A. and M.A.-S.; formal analysis, A.A.; investigation, A.A.; resources, A.A. and M.A.-S.; data curation, M.A.-S.; writing—original draft preparation, M.A.-S.; writing—review and editing, A.A.; visualization, A.A. and M.A.-S.; supervision, A.A.; funding acquisition, M.A.-S. All authors have read and agreed to the published version of the manuscript.

**Funding:** This research received no external funding.

**Data Availability Statement:** On reasonable request, the corresponding author will provide the data that underpin the study's conclusions.

**Acknowledgments:** The employees in the Animal Health and Poultry Department at the Desert Research Center in Egypt and at the Mariut Research Station in Alexandria, Egypt, are acknowledged by the authors.

**Conflicts of Interest:** The authors declare no conflict of interest.

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
