# Peer review of "New Insights on Coding Mutations and mRNA Levels of Candidate Genes Associated with Diarrhea Susceptibility in Baladi Goat"

_agriculture, doi:10.3390/agriculture13010143_

Round 1
Reviewer 1 Report
Manuscript called: New insights on coding mutations and mRNA levels of candidate genes associated with diarrhea susceptibility in Baladi goat describe significance of nucleotide variants and expression profile of 15-th possible candidate genes as biomarkers for diarrhera resistance/susceptibility in Baladi goat. The manuscript meets the requirements of the research work, i.e. is divided into individual chapters. The abstract briefly describes the research work. The introduction briefly describes the breeding and importance of goats, the most frequent mortality of newborn goats, which is associated with diarrhea, its causes and immunity of goats, but also describes the genetic background (genotypic and phenotypic variance, genomics) as well as its impact on economic properties. The material and methodology and procedure are written clearly, comprehensibly and correctly. The used software and statistical analyzes are properly presented. The interpretation of the results is good. The results are interesting and new, which need to be verified in practice, and based on the obtained results, a breeding program should be created. The given links are correct. I recommend this mentioned manuscript for acceptance into the MDPI journal for publication.
Questions:
Line 256-261 How do you explain that? What is it constional on?
Line 406 - infectious diseases- those infectious diseases, can they be divided into some scale, I mean from 1-5 value? 1- means little infectious and 5 very infectious, if so, they can be divided into subgroups, which causes some animals (newborn goats) are more or less infectious - of course, each animal has its own immunity - it can also have a genetic undertone or it is more of a physiological cause or environmental hygiene - how could you explained it?
Also I recommend - English language and style are fine/minor spell check required
Author Response
Reviewer 1
Comment
Manuscript called: New insights on coding mutations and mRNA levels of candidate genes associated with diarrhea susceptibility in Baladi goat describe significance of nucleotide variants and expression profile of 15-th possible candidate genes as biomarkers for diarrhera resistance/susceptibility in Baladi goat. The manuscript meets the requirements of the research work, i.e. is divided into individual chapters. The abstract briefly describes the research work. The introduction briefly describes the breeding and importance of goats, the most frequent mortality of newborn goats, which is associated with diarrhea, its causes and immunity of goats, but also describes the genetic background (genotypic and phenotypic variance, genomics) as well as its impact on economic properties. The material and methodology and procedure are written clearly, comprehensibly and correctly. The used software and statistical analyzes are properly presented. The interpretation of the results is good. The results are interesting and new, which need to be verified in practice, and based on the obtained results, a breeding program should be created. The given links are correct. I recommend this mentioned manuscript for acceptance into the MDPI journal for publication.
Response
We thank reviewer for this positive comment.
Comment
Line 256-261 How do you explain that? What is it constional on?
Response
We thank reviewer for this comment. Using the gene expression profile of the researched genes as an independent variable, a discriminant analysis model was utilized to evaluate the relevance of many variables in order to distinguish between affected and healthy goats as a dependent variable. The goal was to discriminate between diarrheic and healthy kids relied on the mRNA levels of genes under investigation. The interaction between two factors (gene type and diarrhea resistance/susceptibility) and its impact on the gene expression outcomes parameter was evaluated using a univariate general linear model (GLM) with the two-way ANOVA. The statistical analysis is illustrated.
Comment
Line 406 - infectious diseases- those infectious diseases, can they be divided into some scale, I mean from 1-5 value? 1- means little infectious and 5 very infectious, if so, they can be divided into subgroups, which causes some animals (newborn goats) are more or less infectious - of course, each animal has its own immunity - it can also have a genetic undertone or it is more of a physiological cause or environmental hygiene - how could you explained it?
Response
- We thank reviewer for this comment. Detection of diarrhea was based on characters of fecal matter (semisolid to watery). In addition we based on simultaneous recording of temperature, pulse, and respiration rates, color, mucous membranes (Radostits et al., 2007). When the kid has the aforementioned parameters, it is considered healthy (normal fecal matter, temperature, pulse, and respiration rates, mucous membranes).
- Based on clinical observations in neonatal goats, the molecular changes may clarify the diagnosis of diarrhea and offer important hints about how the physiology of the intestines, inflammation, and immunity are related. As a result, This work used real-time PCR and PCR-DNA sequencing to investigate potential genes (TMED1, CALR, FBXW9, HS6ST3, SMURF1, KPNA7, FBXL2, PIN1, S1PR5, ICAM1, EDN1, MAPK11, CSF1R, LRRK1 and CFH) impacting the incidence of diarrhea resistance/susceptibility in Baladi goats through DNA sequencing of these genes alongside real time approach.
- In recent investigations, genome wide association analysis was utilized to identify novel genes specifically for ruminant sensitivity to diarrhea. However, no prior research have looked at the SNPs in these genes and their relationship to goat diarrhea susceptibility.
- The study was conducted on the level of animal response. The main aim was to prove the findings could support the importance of nucleotide variations and the expression pattern of the examined genes as biomarkers for diarrhea resistance/susceptibility and offer a useful management strategy for Baladi goats
- Finally the marked alteration in the gene expression profile (quantitative PCR) of examined inflammatory markers supports a strong inflammatory response that make us these kids are affected by infectious diarrhea.
Comment
Also I recommend - English language and style are fine/minor spell check required
Response
We thank reviewer for this comment. English language and style are fine/minor checked. Better quality image is added.
Reviewer 2 Report
This is an important research about the genetic analysis about diarrheic for goats, and the authors have found some potential genes and markers which may related with diarrheic of goat kids. However, the manuscript still need to improve both in scientific statement and quality of writing. Here are some suggestions.
1. In Materials and Methods part, it should explain why these 15 genes were chosen for the analysis.
2. In 2.2 part, the criteria for defining a diarrheic kid should be more stringent, with exact body temperature and respiration rates...
3. In 2.6 part, the hypothesis should be made for each gene/SNP separately, and the GLM model should be explained in more details, by explaining all the variables in the model.
4. Line 265-271 are not closely related with the purpose of the study, suggest to remove.
5. Some further functional analysis or pathway analysis were suggested to further evaluate the discovered genes.
Author Response
Reviewer 2
Comment
This is an important research about the genetic analysis about diarrheic for goats, and the authors have found some potential genes and markers which may related with diarrheic of goat kids. However, the manuscript still need to improve both in scientific statement and quality of writing. Here are some suggestions.
Response
We thank reviewer for this comment.
Comment
- In Materials and Methods part, it should explain why these 15 genes were chosen for the analysis.
Response
- We thank reviewer for this comment. We are already added a detailed information about the examined genes in the discussion section.
- We are deeply indebted to the reviewer if leave this point according to the authors consideration. Therefore, we prefer adding this part in the discussion section for deciphering the nucleotide sequence variation and alteration in the expression profile in the examined markers in healthy and diarrheic kids.
Comment
- In 2.2 part, the criteria for defining a diarrheic kid should be more stringent, with exact body temperature and respiration rates...
Response
We are grateful to the reviewer for drawing it to our consideration. The exact numbers are added in the newly born kids.
Comment
- In 2.6 part, the hypothesis should be made for each gene/SNP separately, and the GLM model should be explained in more details, by explaining all the variables in the model.
Response
- We thank reviewer for this comment. The overall genes are included in the model to discover which is the most up-regulated or down-regulated gene to understand the regulatory mechanism of each gene in healthy and diarrheic kids (Darwish et al., 2021). Therefore, we can discriminate between healthy and diarrheic kids. Using the gene expression profile of the researched genes as an independent variable, a discriminant analysis model was utilized to evaluate the relevance of many variables in order to distinguish between affected and healthy goats as a dependent variable. The goal was to discriminate between diarrheic and healthy kids relied on the mRNA levels of genes under investigation. The interaction between two factors (gene type and diarrhea resistance/susceptibility) and its impact on the gene expression outcomes parameter was evaluated using a univariate general linear model (GLM) with the two-way ANOVA. The statistical analysis is illustrated.
- Using RFLP and SNP genetic markers in domestic animals, prior research investigated the polymorphism of immune genes in ruminant. Contrarily, we used SNP genetic markers and gene expression to examine gene polymorphism in order to overcome the limitations of previous studies. As a result, in both diarrheal and healthy kids, the examined gene regulation mechanisms are well understood.
Comment
- Line 265-271 are not closely related with the purpose of the study, suggest to remove.
Response
We are grateful to the reviewer for drawing it to our consideration. The paragraph is removed. Consequently, order of the references is changed
Comment
- Some further functional analysis or pathway analysis were suggested to further evaluate the discovered genes.
Response
- We thank reviewer for this comment. Nucleotide sequence variations between the examined genes in the research animals (healthy and diarrheic) and the reference sequences retrieved in GenBank were used to validate all discovered SNPs (Figures S1 to S15). The supplementary figures are added.
- The variants identified in Table 3 are all located within exonic region of studied genes; resulting in coding mutations between healthy and diarrheic kids. We performed amino acid sequence analysis using Mega 4 software. We think it is a functional analysis because when the gene functioning it produce protein.
- Recent studies conducted the genome wide association analysis to target new genes specific for diarrhea susceptibility in livestock; however until now there are no previous studies investigated these genes SNPs and their association with diarrhea susceptibility. With the help of the goat (Capra hircus) gene sequences published in PubMed, our study is the first to indicate this association. Additionally, to the best of our knowledge, no research have previously examined the polymorphism of TMED1, CALR, FBXW9, HS6ST3, SMURF1, KPNA7, FBXL2, PIN1, S1PR5, ICAM1, EDN1, MAPK11, CSF1R, LRRK1, and CFH genes and their relationship to diarrhea in goat.